# SPECTRANET: MULTIVARIATE FORECASTING AND IMPUTATION UNDER DISTRIBUTION SHIFTS AND MISSING DATA

## ABSTRACT

In this work, we tackle two widespread challenges in real applications for time-series forecasting that have been largely understudied: distribution shifts and missing data. We propose `SpectraNet`, a novel multivariate time-series forecasting model that dynamically infers a latent space spectral decomposition to capture current temporal dynamics and correlations on the recent observed history. A Convolution Neural Network maps the learned representation by sequentially mixing its components and refining the output. Our proposed approach can simultaneously produce forecasts and interpolate past observations and can, therefore, greatly simplify production systems by unifying imputation and forecasting tasks into a single model. `SpectraNet` achieves SoTA performance simultaneously on both tasks on five benchmark datasets, compared to forecasting and imputation models, with up to 92% fewer parameters and comparable training times. On settings with up to 80% missing data, `SpectraNet` has average performance improvements of almost 50% over the second-best alternative.

## 1 INTRODUCTION

Multivariate time-series forecasting is an essential task in a wide range of domains. Forecasts are a key input to optimize the production and distribution of goods (Böse et al., 2017), predict healthcare patient outcomes (Chen et al., 2015), plan electricity production (Olivares et al., 2022), build financial portfolios (Emerson et al., 2019), among other examples. Due to its high potential benefits, researchers have dedicated many efforts to improving the capabilities of forecasting models, with breakthroughs in model architectures and performance (Benidis et al., 2022).

The main focus of research in multivariate forecasting has been on accuracy and scalability, to which the present paper contributes. In addition, we identify two widespread challenges for real applications which have been largely understudied: *distribution shifts* and *missing data*.

We refer to *distribution shifts* as changes in the time-series behavior. In particular, we focus on discrepancies in distribution between the train and test data, which can considerably degrade the accuracy (Kuznetsov & Mohri, 2014; Du et al., 2021; Xu et al., 2022; Ivanovic et al., 2022). This has become an increasing problem in recent years with the COVID-19 pandemic, which disrupted all aspects of human activities. *Missing values* is a generalized problem. Some common causes include faulty sensors, the impossibility of gathering data, and misplacement of information. As we demonstrate in our experiments, these challenges hinder the performance of current state-of-the-art (SoTA), limiting their use in applications where these problems are predominant.

In this work, we propose `SpectraNet`, a novel multivariate forecasting model that achieves SoTA performance in benchmark datasets and is also *intrinsically* robust to distribution shifts and extreme cases of missing data. `SpectraNet` achieves its high accuracy and robustness by dynamically inferring a latent vector projected on a temporal basis, a process we name *latent space spectral decomposition* (LSSD). A series of convolution layers then synthesize both the reference window, which is used to infer the latent vectors and the forecast window.

To the best of our knowledge, `SpectraNet` is also the first solution that can simultaneously forecast the future values of a multivariate time series and accurately impute the past missing data.

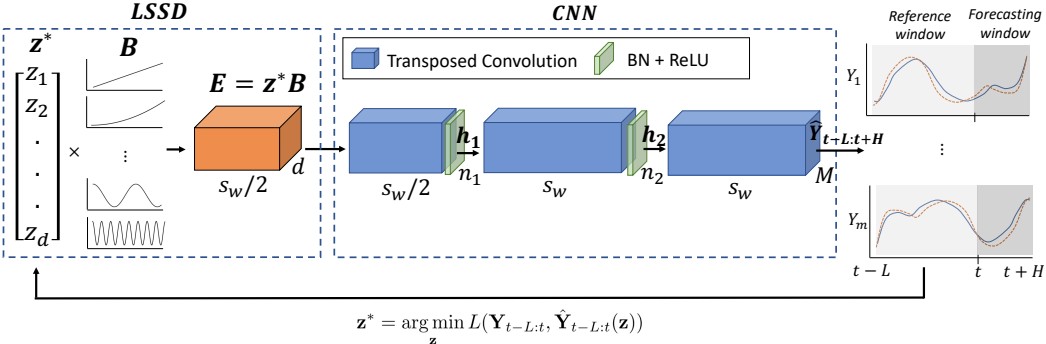

$$\mathbf{z}^* = \arg\min_{\mathbf{z}} L(\mathbf{Y}_{t-L:t}, \hat{\mathbf{Y}}_{t-L:t}(\mathbf{z}))$$

Figure 1: `SpectraNet` architecture. The Latent Space Spectral Decomposition (LSSD) encodes shared temporal dynamics of the target window into Fourier waves and polynomial functions. Latent vector **z** is inferred with Gradient Descent minimizing reconstruction error on the *reference window*. The Convolution Network (CNN) generates the time-series window by sequentially mixing the components of the embedding and refining the output.

In practice, imputation models are first used to fill the missing information for all downstream tasks, including forecasting. `SpectraNet` can greatly simplify production systems by unifying imputation and forecasting tasks into a single model.

The main contributions are:

- **Latent Vector Inference**: methodology to dynamically capture current dynamics of the target time-series into a latent space, replacing parametric encoders.

- **Latent Space Spectral Decomposition:** representation of a multivariate time-series window on a shared latent space with temporal dynamics.

- **SpectraNet**: novel multivariate forecasting model that simultaneously imputes missing data and forecasts future values, with SoTA performance on several benchmark datasets and demonstrated robustness to distribution shifts and missing values. We will make our code publicly available upon acceptance.

The remainder of this paper is structured as follows. Section 2 introduces notation and the problem definition, Section 3 presents our method, Section 4 describes and presents our empirical findings. Finally, Section 5 concludes the paper. The literature review is included in A.1.

## 2 NOTATION AND PROBLEM DEFINITION

We introduce a new notation that we believe is lighter than the standard notation while being intuitive and formally correct. Let $\mathbf{Y} \in \mathbb{R}^{M \times T}$ be a multivariate time-series with $M$ features and $T$ timestamps. Let $\mathbf{Y}_{a:b} \in \mathbb{R}^{M \times (b-a)}$ be the observed values for the interval $[a, b)$, that is, $\mathbf{Y}_{0:t}$ is the set of $t$ observations of $\mathbf{Y}$ from timestamp 0 to timestamp $t - 1$ while $\mathbf{Y}_{t:t+H}$ is the set of $H$ observations of $\mathbf{Y}$ from timestamp $t$ to timestamp $t + H - 1$. Let $y_{m,t} \in \mathbb{R}$ be the value of feature $m$ at timestamp $t$.

In this work we consider the multivariate point forecasting task, which consists of predicting the future values of a multivariate time-series sequence based on past observations. The main task of a model $F_{\Theta}$ with parameters $\Theta$ at a timestamp $t$, is to produce forecasts for the future $H$ values, denoted by $\hat{\mathbf{Y}}_{t:t+H}$, based on the previous history $\mathbf{Y}_{0:t}$.

$$\hat{\mathbf{Y}}_{t:t+H} = F_{\Theta}(\mathbf{Y}_{0:t}) \qquad (1)$$

For the imputation task, to impute a missing value $y_{m,t}$, models are not constrained to only use past observations. Moreover, they are evaluated on how well they approximate only the missing values. We evaluate the performance with two common metrics used in the literature, mean squared error (MSE) and mean absolute error (MAE), given by equation 2 (Hyndman & Athanasopoulos, 2018).

$$\text{MSE} = \frac{1}{MH} \sum_{h=0}^{H-1} \sum_{m=1}^{M} (y_{m,t+h} - \hat{y}_{m,t+h})^2 \qquad \text{MAE} = \frac{1}{MH} \sum_{h=0}^{H-1} \sum_{m=1}^{M} |y_{m,t+h} - \hat{y}_{m,t+h}|. \quad (2)$$

## 3  SPECTRANET

We start the description of our approach with a general outline of the model and explain each major component in detail in the following subsections. The overall architecture is illustrated in Figure 1. `SpectraNet` is a top-down model that *generates* a multivariate time-series window of fixed size $s_w = L + H$, where $L$ is the length of the *reference window* and $H$ is the forecast horizon, from a latent vector $\mathbf{z} \in \mathbb{R}^d$. To produce the forecasts at timestamp $t$, the model first *infers* the optimal latent vector on the *reference window*, consisting of the last $L$ values, $\mathbf{Y}_{t-L:t}$, by minimizing the *reconstruction error*. This inference step is the main difference between our approach to existing models, which map the input into an embedding or latent space using an encoder network. The model generates the full time-series window $s_w$, which includes the *forecast window* $\mathbf{Y}_{t:t+H}$, with a *spectral decomposition* and a Convolutional Neural Network (CNN). The main steps of `SpectraNet` are given by

$$\mathbf{z}^* = \arg\min_{\mathbf{z}} L(\mathbf{Y}_{t-L:t}, \hat{\mathbf{Y}}_{t-L:t}(\mathbf{z})) \qquad (3)$$

where $\mathbf{z}^*$ is the *inferred* latent vector, $L$ is a reconstruction error metric, and $\hat{\mathbf{Y}}_{t-L:t}(\mathbf{z})$ is given by,

$$\mathbf{E} = \text{LSSD}(\mathbf{z}, \mathbf{B}) \qquad (4)$$

$$\hat{\mathbf{Y}}_{t-L:t}, \hat{\mathbf{Y}}_{t:t+H} = \text{CNN}_\Theta(\mathbf{E}) \qquad (5)$$

where LSSD (*latent space spectral decomposition*) is a basis expansion operation of $\mathbf{z}$ over the predefined temporal basis $\mathbf{B}$ to produce a temporal embedding $\mathbf{E} \in \mathbb{R}^{d \times d_t}$, and CNN is a Top-Down Convolutional Neural Network with learnable parameters $\Theta$. The CNN simultaneously produces both the *reconstruction* of the past reference window $\hat{\mathbf{Y}}_{t-L:t}$, used to find the optimal latent vector for the full window and the forecast $\hat{\mathbf{Y}}_{t:t+H}$.

### 3.1  LATENT VECTOR INFERENCE

The proposed latent vector inference is based on the Alternating back-propagation algorithm (ABP) (Han et al., 2017) for training generative models without encoders. A single generator (decoder) architecture is trained by maximizing the observed likelihood directly. To achieve this, at every step, ABP samples latent vectors from the posterior distribution $P(\mathbf{Z}|\mathbf{Y})$ using MCMC methods such as Langevin dynamics. Generative models trained with ABP demonstrated superior performance in recovering missing segments of images and videos over Variational Autoencoders (VAE) and Generative Adversarial Networks (GAN). To our knowledge, `SpectraNet` is the first approach that uses and extends the latent vector inference principle of ABP for time-series forecasting.

We reformulate the posterior distribution sampling of the latent vectors as a non-convex minimization problem presented in 3, which aims to minimize the generator's reconstruction error on the reference window $\mathbf{Y}_{t-L:t}$. We use the mean square error (MSE) as the reconstruction loss since it is differentiable, fast to compute and theoretically founded. Our method corresponds to the maximum a posteriori estimation (MAP) in the ABP framework. Appendix A.2 formally presents the ABP algorithm and the connections to our method.

To solve the reconstruction loss minimization problem, we use gradient-based methods. In particular, we rely on gradient descent (GD), randomly initializing the latent vector with independent and identically distributed Gaussian distributions and fixing the learning rate and the number of iterations as hyperparameters. Figure 2 demonstrates how `SpectraNet`'s output evolves during the inference of $\mathbf{z}$, adapting to current behaviors on the *reference* window. Section 3.4 presents how `SpectraNet` is trained by alternating the inference and parameter learning steps.

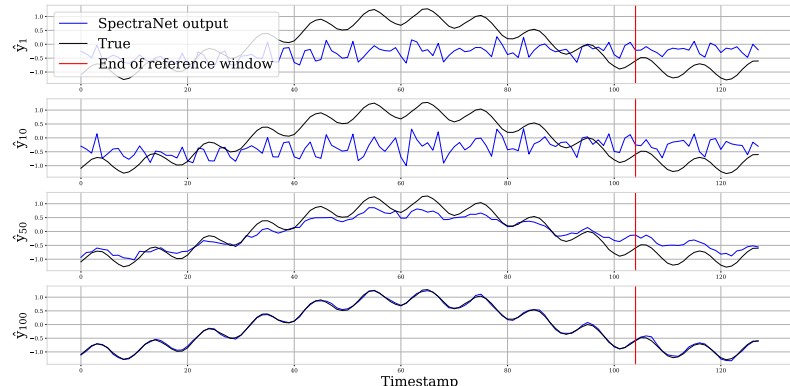

Figure 2: `SpectraNet`'s output evolution during latent vector inference with Gradient Descent. The model maps the latent vector to the complete window, including both *reference* (of size 104) and *forecasting* windows (of size 24), using only information from the former. The temporal basis **B** imposes strict dependencies between both windows. This inference process allows `SpectraNet`to dynamically adapt to new behaviours and forecast with missing data.

The main advantage of inferring the latent vector is the robustness to handle both distribution shifts and missing values. During this step, the model will infer the latent vector that best fits the current dynamics on the reference window, even if they follow an unseen behavior during training. The ability to dynamically adjust the temporal embedding for each window gives `SpectraNet` more robustness to changes in distribution, as shown in Figure 3. To deal with missing data, the reconstruction loss is masked. Following the notation in (Tashiro et al., 2021) , let $M \in \{0,1\}^{M \times T}$ be the observation mask which indicates the availability of data. The latent vectors are inferred following

$$\mathbf{z}^* = \arg \min_{\mathbf{z}} L(\mathbf{M}_{t-L:t} \circ \mathbf{Y}_{t-L:t}, \mathbf{M}_{t-L:t} \circ \hat{\mathbf{Y}}_{t-L:t}(\mathbf{z})) \tag{6}$$

where $\circ$ is the element-wise matrix multiplication. As demonstrated in the experimental section results on Tables 1, 2 and 4, and in congruity with previous studies such as (Han et al., 2017; Pang et al., 2020), inferring latent vectors provides superior robustness to missing data.

### 3.2 LATENT SPACE SPECTRAL DECOMPOSITION

The second component of `SpectraNet` is the mapping from the latent vectors **z** into the temporal embedding **E**. Each element of **z** corresponds to the coefficient of one element of the temporal basis $\mathbf{B} \in \mathbb{R}^{d \times d_t}$, where $d$ is the number of elements and $d_t$ is the temporal length. The $i$-th row of **E** is given by,

$$\mathbf{E}_{\mathbf{i},:} = z_i^* \mathbf{B}_{\mathbf{i},:} \tag{7}$$

Each element of the basis consists of a predefined template function. Similarly to previous work including (Oreshkin et al., 2019; Engel et al., 2020; Shan et al., 2022), the basis includes patterns commonly found in time series: trends, represented by polynomial functions, and seasonal, represented by harmonic functions. The final basis matrix **B** is the row-wise concatenation of the three following matrices.

$$\begin{aligned}
\mathbf{B}_{i,t}^{trd} &= t^i &&, \text{for } i \in \{0, ..., p\}, t \in \{0, ..., d_t\} \\
\mathbf{B}_{i,t}^{cos} &= \cos(2\pi i t) &&, \text{for } i \in \{1, ..., \frac{s_w}{2}\}, t \in \{0, ..., d_t\} \\
\mathbf{B}_{i,t}^{sin} &= \sin(2\pi i t) &&, \text{for } i \in \{1, ..., \frac{s_w}{2}\}, t \in \{0, ..., d_t\}
\end{aligned} \tag{8}$$

where $p$ is a hyperparameter that controls the max degree of the polynomial basis. The final size of the basis $d$ is equal to $s_w + p + 1$. The temporal embedding **E** corresponds to a *latent space spectral decomposition* that encodes shared temporal dynamics of all features in the target window, as the latent vector $\mathbf{z}^*$ selects the relevant trend and frequency bands. Another crucial reason behind using a predefined basis is to impose strict temporal dependencies between the reference and forecasting

windows. While inferring the latent vector, the forecasting window does not provide information (gradients). If all the entries of $\mathbf{E}$ are freely inferred the last values of the temporal embedding that determines the forecasting window cannot be optimized.

### 3.3 TOP-DOWN CONVOLUTION NETWORK

SpectraNet uses a Top-Down Convolution Network (CNN) as decoder, which produces the final forecast and reconstruction of the reference window $\hat{\mathbf{Y}}_{t-L:t+H}$ from the temporal embedding $\mathbf{E}$. The full CNN architecture is presented in Figure 1, and can be formalized as:

$$
\begin{aligned}
\mathbf{h_1} &= \text{ReLU}(\text{BN}(\text{Transposed Convolution}(\mathbf{E}))) \\
\mathbf{h_2} &= \text{ReLU}(\text{BN}(\text{Transposed Convolution}(\mathbf{h_1}))) \\
\hat{\mathbf{Y}}_{t-L:t+H} &= \text{Transposed Convolution}(\mathbf{h_2})
\end{aligned}
\tag{9}
$$

where Transposed Convolution is a transpose convolutional layer on the temporal dimension, BN is a batch normalization layer, and ReLU activations introduce non-linearity. The number of filters at each layer, kernel size, and stride are given in Appendix A.5. The convolutional filters are not *causal* as in a Temporal Convolution Network (TCN) (Bai et al., 2018). Instead, SpectraNet's *causality*, that allows the model to forecast future data using past observations, comes from inferring the latent vector $\mathbf{z}^*$ with the reference window.

The first layers of the CNN learn a common representation for all features from the temporal embedding $\mathbf{E}$. Additionally, the second layer *refines* the temporal resolution to the final size of the window $s_w$. The last layer produces the final output for all $M$ features from $\mathbf{h_2}$. While equation 9 presents the default configuration for SpectraNet used in the experiments, additional layers can be added to increase the expressivity. Appendix A.10 presents ablation studies on the number of layers.

### 3.4 TRAINING PROCEDURE

Each training iteration consists of two steps: the *inference step* and the *learning step*. During the *inference step*, the optimal latent vector $\mathbf{z}^*$ is inferred using Gradient Descent, solving the optimization problem given in equation 3 with the current parameters $\Theta$. During the *learning step*, the latent vector is used as the input to the model, and the parameters $\Theta$ are updated using ADAM optimizer (Kingma & Ba, 2014) and MSE loss. For each iteration, we sample a small batch (with replacement) of multivariate time-series windows $\mathbf{Y}_{t:t+s_w}$ from the training data, each starting at a random timestamp $t$. Each sampled observation is first *normalized* with the mean and standard deviation on the *reference* window, to decouple the scale and patterns (the output is scaled back before evaluation). The full training procedure is presented in Appendix A.6.

One potential drawback of inferring the latent vectors is the computational cost. We tackle this in several ways. First, by solving the optimization problem using GD, we can rely on automatic differentiation libraries such as PyTorch to efficiently compute the gradients. Second, backpropagation on CNN is parallelizable in GPU since it does not require sequential computation. The forecasts are independent between windows so they can be computed in parallel, unlike RNN models, which must produce the forecasts sequentially. Finally, during training, we *persist* the optimal latent vectors to future iterations on the same window. For a given observation $t$, the final latent vector $\mathbf{z_t^*}$ is stored and used as a warm start when observation $t$ is sampled again, considerably reducing the number of iterations during the *inference step*. We discuss training times and memory complexity in section 4.

### 3.5 ROBUSTNESS TO MISSING VALUES AND DISTRIBUTIONAL SHIFTS

We first test SpectraNet's robustness on synthetic data, which we name **Simulated7**. This dataset consists of seven time series of length 20,000, each generated as the sum of two cosines with random frequencies and small Gaussian noise. We inject missing values on both train and test sets, following the procedure described in section 4. For the distribution shifts experiments, we modify the test set with two common distribution shifts: changes in *trend* and in *magnitude*. Figure 3 presents the forecasts for SpectraNet for the three settings considered. We include the first feature in this figure, and we show the complete data in Appendix A.7. As seen in panel (a), our method can accurately forecast the ground truth even with 80% of missing values, a challenging setting since

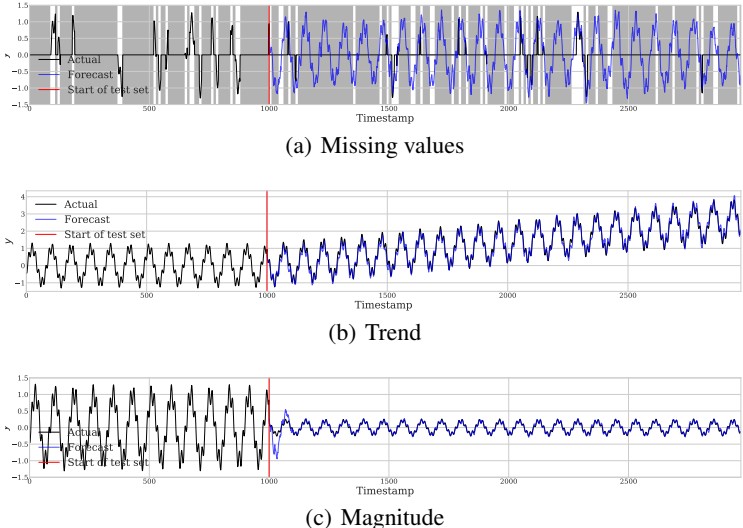

(a) Missing values

(b) Trend

(c) Magnitude

Figure 3: Forecasts for the first feature of **Simulated7** dataset using `SpectraNet` with (a) 80% of missing data (missing regions in grey), (b) change in trend, and (c) change in magnitude. Forecasts are produced every 24 timestamps in a rolling window strategy.

intervals where consecutive timestamps are available are short. This setting also demonstrates how `SpectraNet` can accurately learn correlations between features, as in many *reference* windows only few features contain information. Panels (b) and (c) demonstrate how `SpectraNet` can dynamically adapt to new data behaviors, even if the model never observed them during training. Table 1 presents a quantitative comparison.

Table 1: Forecasting accuracy on **Simulated7** dataset with missing values and distribution shifts between the train and test sets, forecasting horizon of 24 timestamps. Lower scores are better. Metrics are averaged over five runs, best model highlighted in bold.

| | $p_o$ | SpectraNet MSE | SpectraNet MAE | N-HiTS MSE | N-HiTS MAE | Informer MSE | Informer MAE | Autoformer MSE | Autoformer MAE | StemGNN MSE | StemGNN MAE | RNN MSE | RNN MAE |
|---|---|---|---|---|---|---|---|---|---|---|---|---|---|
| Normal | 0.0 | 0.004 | 0.041 | **0.001** | **0.015** | 0.008 | 0.079 | 0.017 | 0.096 | 0.036 | 0.122 | 0.001 | 0.022 |
| Trend | 0.0 | **0.025** | **0.130** | 0.427 | 0.443 | 2.821 | 1.410 | 0.060 | 0.19 | 1.354 | 0.952 | 1.214 | 0.859 |
| Magnitude | 0.0 | **0.005** | **0.038** | 0.009 | 0.059 | 0.020 | 0.094 | 0.043 | 0.168 | 0.060 | 0.192 | 0.089 | 0.248 |
| M. Values | 0.2 | **0.009** | **0.081** | 0.012 | 0.176 | 0.071 | 0.237 | 0.319 | 0.430 | 0.097 | 0.266 | 0.069 | 0.148 |
| M. Values | 0.4 | **0.011** | **0.073** | 0.022 | 0.213 | 0.095 | 0.242 | 0.591 | 0.612 | 0.181 | 0.385 | 0.172 | 0.308 |
| M. Values | 0.6 | **0.018** | **0.100** | 0.133 | 0.338 | 0.231 | 0.380 | 0.752 | 0.994 | 0.362 | 0.488 | 0.357 | 0.462 |
| M. Values | 0.8 | **0.023** | **0.127** | 0.365 | 0.439 | 0.372 | 0.478 | 0.925 | 1.106 | 0.436 | 0.614 | 0.500 | 0.571 |

In Appendix A.9 we demonstrate how `SpectraNet` is also robust to changes over time of the *missing data regime* (how much or for how long missing data occurs). While `SpectraNet` is intrinsically robust to missing data, current SoTA models require either modifications to the architecture or an interpolation model to first impute the missing values. Another advantage of our method is that it cannot only forecast directly on the raw data, but by generating the entire window, it is simultaneously interpolating the past missing data and forecasting the future values.

# 4 EXPERIMENTAL RESULTS

We base our experimental setting, benchmark datasets, train/validation/test splits, hyperparameter tuning, and data processing on previous works on multivariate forecasting (Zhou et al., 2021; Wu et al., 2021; Challu et al., 2022a).

## 4.1 Datasets

We evaluate our model on synthetic data and five popular benchmark datasets commonly used in the forecasting literature, comprising various applications and domains. All datasets are normalized with the mean and standard deviation computed on the train set. We split each dataset into train, validation, and test sets, following different proportions based on the number of timestamps. We present summary statistics for each dataset in Appendix A.3.

The **Influenza-like illness (ILI)** dataset contains the weekly proportion of patients with influenza-like symptoms in the US, reported by the Centers for Disease Control and Prevention, from 2002 to 2021. **Exchange** reports the daily exchange rates of eight currencies relative to the US dollar from 1990 to 2016. The **Solar** dataset contains the hourly photo-voltaic production of 32 solar stations in Wisconsin during 2016. Finally, the **Electricity Transformer Temperature (ETTm2)** dataset is a stream of eight sensor measurements of an electricity transformer in China from July 2016 to July 2018. **Weather** contains 21 meteorological conditions recorded at the German Max Planck Biogeochemistry Institute in 2021.

## 4.2 Training and Evaluation setup

We train models on the training set comprising the earliest history of each dataset. We select the optimal hyperparameters for all models (including the baselines to ensure fair comparisons) on each dataset and occlusion percentage based on the performance (measured by MSE) on the validation set, using a Bayesian optimization algorithm, HYPEROPT (Bergstra et al., 2011). For `SpectraNet` we tune the number of convolutional filters, temporal size of the latent vectors, window normalization, and optimization hyperparameters for both the inferencial and learning steps. The complete list of hyperparameters is included in Appendix A.5. For the main results, we use a multi-step forecast with a horizon of size 24, and forecasts are produced in a rolling window strategy. We repeat the experiment five times with different random seeds and report average performances. We run the experiments on an AWS `g3.4xlarge` EC2 instance with NVIDIA Tesla M60 GPUs.

We compare our proposed model against several univariate and multivariate state-of-the-art baselines based on different architectures: Transformers, feed-forward networks (MLP), Graph Neural Network (GNN), and Recurrent Neural Network (RNN). For Transformer-based models we include the Informer (Zhou et al., 2021) and Autoformer (Wu et al., 2021), two recent models for multi-step long-horizon forecasting; for MLP we consider the univariate N-HiTS (Challu et al., 2022a) and N-BEATS (Oreshkin et al., 2019) models; for GNN models we include the StemGNN (Cao et al., 2020), and lastly we include a multi-step univariate RNN with dilations (Chang et al., 2017).

## 4.3 Missing data setup

We present a new experimental setting for testing the forecasting models' robustness to missing data. We parameterize the experiments with two parameters: the size of missing segments $s$ and the probability that each segment is missing with $p_o$. First, the original time series is divided into disjoint segments of length $s$. Second, each feature $m$ in each segment is occluded with probability $p_o$. We repeat the experiment with different probabilities $p_o$: 0% (no missing values), 20%, 40%, 60%, and 80%, to test models' performance under different proportions of missing values. The size of segments $s$ is fixed at ten timestamps for ILI and at 100 for all other datasets. Figure 3 shows an example of the setting for the **Simulated7** dataset with 80% missing data (occluded data in grey).

## 4.4 Key results

**Forecasting accuracy**. Table 2 presents the forecasting accuracy. First, `SpectraNet` achieved SoTA performance even on full data, showing the advantages of our method are not limited to the robustness to missing data and distribution shifts. `SpectraNet` outperformed N-HiTS/N-BEATS in two of the five datasets and placed in the Top-3 models across all datasets, outperforming Transformer-based models, StemGNN, and DilRNN. `SpectraNet` also achieves the best performance among all multivariate models.

The superior robustness of `SpectraNet` to missing values is evident in all datasets and proportions. In datasets with strong seasonalities, such as **Simulated7** and **Solar**, the accuracy of

Table 2: Main forecasting accuracy results on benchmark datasets with different proportion of missing values ($p_o$), forecasting horizon of 24 timestamps, lower scores are better. Metrics are averaged over five runs, best model highlighted in bold.

| | $p_o$ | SpectraNet MSE | MAE | N-HiTS/NBEATS MSE | MAE | Informer MSE | MAE | Autoformer MSE | MAE | StemGNN MSE | MAE | RNN MSE | MAE |
|---|---|---|---|---|---|---|---|---|---|---|---|---|---|
| ILI | 0.0 | **0.724** | **0.557** | 1.379 | 0.762 | 4.265 | 1.329 | 2.249 | 0.967 | 4.013 | 1.437 | 3.852 | 1.340 |
| | 0.2 | **1.153** | **0.662** | 2.028 | 0.933 | 3.624 | 1.127 | 3.250 | 1.292 | 4.372 | 1.503 | 3.647 | 1.321 |
| | 0.4 | **1.646** | **0.793** | 3.248 | 1.134 | 3.908 | 1.351 | 4.308 | 1.419 | 4.752 | 1.694 | 4.360 | 1.455 |
| | 0.6 | **2.453** | **1.012** | 3.871 | 1.257 | 3.991 | 1.277 | 4.571 | 1.515 | 5.170 | 1.614 | 5.561 | 1.614 |
| | 0.8 | **3.136** | **1.139** | 5.102 | 1.451 | 4.180 | 1.363 | 4.745 | 1.536 | 5.337 | 1.836 | 5.063 | 1.622 |
| Exchange | 0.0 | 0.048 | 0.157 | **0.031** | **0.102** | 0.472 | 0.534 | 0.049 | 0.167 | 0.102 | 0.251 | 0.086 | 0.210 |
| | 0.2 | **0.091** | **0.223** | 0.295 | 0.342 | 1.013 | 0.772 | 0.758 | 0.523 | 1.021 | 0.599 | 0.871 | 0.529 |
| | 0.4 | **0.158** | **0.292** | 0.321 | 0.534 | 1.162 | 0.905 | 0.782 | 0.606 | 0.826 | 0.608 | 0.694 | 0.550 |
| | 0.6 | **0.367** | **0.418** | 0.549 | 0.745 | 1.564 | 0.932 | 1.346 | 0.849 | 1.991 | 1.002 | 1.432 | 0.872 |
| | 0.8 | **1.125** | **0.785** | 1.792 | 1.188 | 2.530 | 1.281 | 2.520 | 1.231 | 2.778 | 1.318 | 2.791 | 1.312 |
| Solar | 0.0 | **0.007** | **0.054** | 0.008 | 0.061 | 0.011 | 0.065 | 0.018 | 0.095 | 0.015 | 0.077 | 0.012 | 0.060 |
| | 0.2 | **0.012** | **0.058** | 0.016 | 0.063 | 0.016 | 0.083 | 0.025 | 0.116 | 0.016 | 0.084 | 0.015 | 0.065 |
| | 0.4 | **0.012** | **0.062** | 0.017 | 0.068 | 0.020 | 0.102 | 0.023 | 0.118 | 0.023 | 0.103 | 0.017 | 0.064 |
| | 0.6 | **0.013** | **0.068** | 0.022 | 0.075 | 0.021 | 0.107 | 0.027 | 0.130 | 0.031 | 0.115 | 0.026 | 0.083 |
| | 0.8 | **0.014** | **0.072** | 0.025 | 0.089 | 0.035 | 0.156 | 0.036 | 0.127 | 0.065 | 0.192 | 0.028 | 0.113 |
| ETTm2 | 0.0 | 0.136 | 0.212 | **0.116** | **0.203** | 0.366 | 0.462 | 0.171 | 0.275 | 0.154 | 0.273 | 0.179 | 0.238 |
| | 0.2 | **0.155** | **0.260** | 0.656 | 0.420 | 0.895 | 0.704 | 0.512 | 0.450 | 0.963 | 0.581 | 0.574 | 0.421 |
| | 0.4 | **0.228** | **0.316** | 1.191 | 0.637 | 0.923 | 0.679 | 1.078 | 0.683 | 1.542 | 0.795 | 0.807 | 0.584 |
| | 0.6 | **0.500** | **0.442** | 1.976 | 0.918 | 1.784 | 0.999 | 1.643 | 0.891 | 2.151 | 0.978 | 1.163 | 0.741 |
| | 0.8 | **0.725** | **0.528** | 2.019 | 1.230 | 1.982 | 1.115 | 1.992 | 1.062 | 2.595 | 1.162 | 2.085 | 1.007 |
| Weather | 0.0 | 0.112 | 0.193 | 0.109 | **0.127** | 0.218 | 0.283 | 0.186 | 0.263 | **0.103** | 0.152 | 0.124 | 0.201 |
| | 0.2 | **0.154** | 0.218 | 0.181 | **0.205** | 0.287 | 0.350 | 0.324 | 0.389 | 0.179 | 0.223 | 0.189 | 0.240 |
| | 0.4 | **0.205** | **0.269** | 0.264 | 0.279 | 0.320 | 0.385 | 3.105 | 1.456 | 0.296 | 0.311 | 0.291 | 0.325 |
| | 0.6 | **0.329** | **0.387** | 0.420 | 0.414 | 0.761 | 0.628 | 3.772 | 1.660 | 0.495 | 0.569 | 0.435 | 0.433 |
| | 0.8 | **0.439** | **0.463** | 0.517 | 0.503 | 0.915 | 0.889 | 4.204 | 1.859 | 0.760 | 0.702 | 0.480 | 0.48 |

SpectraNet marginally degrades with even 80% of missing values. The relative performance of our method improves with the proportion of missing data. For example, for 80%, the average MSE across datasets is 48% lower than the N-HiTS and 65% lower than the Autoformer. The improvement on **ILI** with complete data of almost 50% against the second best model is explained by the *distribution shift* between the train and test set, where the latter presents larger spikes and a clear positive trend on some features. Appendix A.8 presents a plot of the ILI dataset.

**Imputation accuracy**. Table 3 presents the results for the imputation task. We include a SoTA imputation diffusion model, CSDI (Tashiro et al., 2021), and three simple baselines: (i) imputation with the mean of each feature, (ii) imputation with the last available value (Naive), and (iii) linear interpolation between past and future available values. SpectraNet consistently achieves the best performance across all datasets. CSDI outperformes all simpler baselines on Simulated7 and Solar, but its performance degrades on datasets with distribution shifts such as Simulated7-Trend and ILI.

Table 3: Imputation accuracy on the test set for SpectraNet and baselines with different proportion of missing data ($p_o$). Lower scores are better, best model highlighted in bold.

| | $p_o$ | SpectraNet MSE | MAE | CSDI MSE | MAE | Mean MSE | MAE | Naive MSE | MAE | Linear MSE | MAE |
|---|---|---|---|---|---|---|---|---|---|---|---|
| Simulated7 | 0.2 | **0.008** | **0.070** | 0.062 | 0.184 | 0.535 | 0.642 | 0.188 | 0.344 | 0.069 | 0.213 |
| | 0.6 | **0.022** | **0.116** | 0.218 | 0.334 | 0.551 | 0.656 | 0.468 | 0.518 | 0.225 | 0.341 |
| Simulated7-Trend | 0.2 | **0.029** | **0.136** | 0.546 | 0.536 | 2.067 | 1.196 | 0.238 | 0.364 | 0.083 | 0.229 |
| | 0.6 | **0.063** | **0.185** | 0.757 | 1.009 | 1.862 | 1.127 | 0.470 | 0.518 | 0.228 | 0.343 |
| Simulated7-Magnitude | 0.2 | **0.005** | **0.039** | 0.009 | 0.053 | 0.033 | 0.139 | 0.024 | 0.087 | 0.010 | 0.055 |
| | 0.6 | **0.011** | **0.053** | 0.051 | 0.108 | 0.036 | 0.144 | 0.064 | 0.146 | 0.033 | 0.094 |
| Solar | 0.2 | **0.004** | **0.042** | 0.022 | 0.096 | 0.038 | 0.182 | 0.052 | 0.139 | 0.032 | 0.116 |
| | 0.6 | **0.006** | **0.048** | 0.034 | 0.103 | 0.038 | 0.182 | 0.056 | 0.145 | 0.039 | 0.128 |
| Exchange | 0.2 | **0.018** | 0.058 | 0.045 | 0.093 | 1.560 | 1.002 | 0.071 | 0.105 | 0.023 | **0.055** |
| | 0.6 | **0.035** | **0.084** | 0.101 | 0.196 | 1.573 | 1.010 | 0.099 | 0.138 | 0.042 | 0.089 |
| ILI | 0.2 | **0.592** | **0.603** | 2.153 | 0.903 | 4.085 | 1.412 | 1.878 | 0.871 | 0.630 | 0.642 |
| | 0.6 | **2.568** | **1.349** | 5.131 | 1.508 | 7.351 | 1.880 | 7.840 | 1.895 | 7.245 | 1.605 |

**Memory and time complexity**. We compare the training time and memory usage (as the number of learnable parameters) of SpectraNet and baseline models as a function of the input size on the ETTm2 dataset in Figure 4, using the optimal hyperparameters. Panel (a) shows our method has

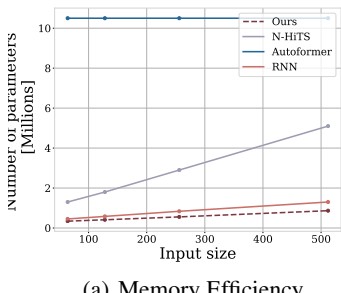
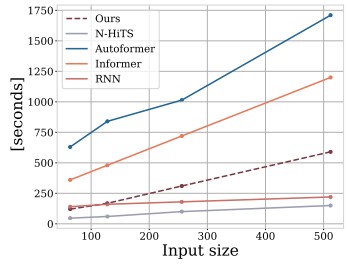

(a) Memory Efficiency        (b) Time Efficiency

Figure 4: Memory efficiency and train time analysis on ETTm2. Memory efficiency measured as the number of parameters, train time includes the complete training procedure. We use the best hyperparameter configuration for each model, based on model accuracy.

the lowest memory footprint, with up to **85%** fewer parameters than the N-HiTS (the second best alternative in most datasets) and **92%** than the Autoformer. Despite the fact that `SpectraNet` performs the inference step in each iteration, thanks to the improvements discussed in section 3 and the reduced number of parameters, training times are comparable to other baseline models.

**Ablation studies**. Finally, we test the contribution of the two major components proposed in this work, the dynamic inference of latent vectors and the LSSD. We compare `SpectraNet` against two versions without these components. In `SpectraNet`$_1$ we add a CNN encoder to map inputs to a standard embedding, and in `SpectraNet`$_2$ we keep the inference step but remove the LSSD. Table 4 presents the average performance across the five benchmark datasets. As expected, `SpectraNet`$_1$ with the parametric encoder is not robust to missing data, with similar performance to other SoTA models. On top of improving the performance over `SpectraNet`$_2$, the LSSD reduces the number of parameters by 70% since fewer layers are needed to generate the complete window.

| $p_o$ | SpectraNet$_1$ MSE | SpectraNet$_1$ MAE | SpectraNet$_2$ MSE | SpectraNet$_2$ MAE | SpectraNet MSE | SpectraNet MAE |
|---|---|---|---|---|---|---|
| 0.0 | 0.691 | 0.461 | 0.237 | 0.283 | **0.213** | **0.251** |
| 0.2 | 0.839 | 0.623 | 0.382 | 0.328 | **0.310** | **0.284** |
| 0.4 | 1.412 | 0.666 | 0.492 | 0.427 | **0.449** | **0.341** |
| 0.6 | 1.952 | 0.974 | 0.916 | 0.620 | **0.752** | **0.467** |
| 0.8 | 2.541 | 1.082 | 1.547 | 0.859 | **1.126** | **0.579** |

Table 4: Average forecasting accuracy across five benchmark datasets for `SpectraNet` and two versions without main components. Lower scores are better, best highlighted in bold.

## 5 DISCUSSIONS AND CONCLUSION

This work proposes a novel multivariate time-series forecasting model that uses a Top-Down CNN to generate time-series windows from a novel latent space spectral decomposition. It dynamically infers the latent vectors that best explain the current dynamics, considerably improving the robustness to distribution shifts and missing data. We compare the accuracy of our method with SoTA models based on several architectures for forecasting and interpolation tasks. We demonstrate `SpectraNet` does not only achieve SoTA on benchmark datasets but can also produce accurate forecasts under some forms of distribution shifts and extreme cases of missing data.

Our experiments provide evidence that, as expected, the performance of current SoTA forecasting models significantly degrades under the presence of missing data and distribution shifts. These challenges are commonly present in high stake settings such as healthcare (for instance, cases with up to 80% missing data are common) and are becoming more predominant in many domains with the recent global events such as the COVID-19 pandemic. Designing robust algorithms and solutions that tackle these challenges can significantly improve their adoption and increase their benefits.

We believe that dynamically inferring latent vectors can have multiple other applications in time-series forecasting. For instance, it can have applications in transfer learning and few-shot learning. A pre-trained decoder could forecast completely unseen time series, while the inference step will allow the model to adapt to different temporal patterns. Moreover, the inference step can be adapted to produce multiple samples and calibrated to learn the target time series distribution.

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

# A APPENDIX

## A.1 RELATED WORK

**Time-series forecasting.** Time-series forecasting has long been an active area of research in academia and industry due to its broad range of high-impact applications. The latest developments in machine learning have been combined with classical methods or used to develop new models (Januschowski et al., 2020) which have been successful in recent large-scale forecasting competitions (Makridakis et al., 2020; 2022). The earlier breakthroughs in modeling time-series with Deep-learning (Januschowski et al., 2018) include the widely used Recurrent Neural Networks (RNN) (Chang et al., 2017; Salinas et al., 2020; 2019) and Temporal Convolution Networks (TCN) (Bai et al., 2018). The success of Transformers (Vaswani et al., 2017) in sequential data, such as natural language processing (NLP) and audio processing, inspired many recent models with attention mechanisms. The Informer (Zhou et al., 2021) introduces a Prob-sparse self-attention to reduce the quadratic complexity of vanilla Transformers; the Autoformer (Wu et al., 2021) proposed a decomposition architecture in trend and seasonal components, and the Auto-correlation mechanism.

Simpler approaches based on feed-forward networks have also shown remarkable performance. The N-BEATS model (Oreshkin et al., 2019) uses a deep stack of fully-connected layers to decompose the forecast into both learnable and predefined basis functions; the NBEATSx extension incorporates interpretable exogenous variable decomposition (Olivares et al., 2022); the N-HiTS model (Challu et al., 2022a) generalizes the N-BEATS by introducing mixed-data sampling and hierarchial interpolation to decompose the forecast in different frequencies. Finally, Graph Neural Networks (GNN) has been used to incorporate complex relations between a large number of time series, including the GraphWaveNet (Wu et al., 2019), and StemGNN (Cao et al., 2020) models.

**Time-series imputation.** The standard practice to handle missing data is filling the missing information, a process called *interpolation*. Simple interpolation alternatives include replacing missing values with zeros, the mean, most-recent value (naive), and linear interpolation. Most recent Deep-learning approaches consist of Generative Adversarial Networks (GANs) and RNN-based architectures. Some notable examples are E2gan (Luo et al., 2019), Brits (Cao et al., 2018), and NAOMI (Liu et al., 2019). More recent approaches include the CDSI (Tashiro et al., 2021) model, a score-based diffusion auto-regressive architecture that produces a distribution for the imputed values.

**Alternating back-propagation**. The method we propose to infer latent vectors is inspired by the Alternating back-propagation algorithm (ABP) for Generative models (Han et al., 2017). The key idea of this algorithm is to sample latent vectors from the posterior distribution with MCMC methods and train a Generative model by maximizing the observed likelihood directly. Generative models trained with ABP do not need an encoder, such as Variational Autoencoders (VAE), or Discriminator networks, such as GANs. More recent work extend the original architecture with energy-based models for CV (Pang et al., 2020), LSTM networks for text generation (Pang et al., 2021), and a hierarchical latent space for anomaly detection (Challu et al., 2022b).

## A.2 ALTERNATING BACK-PROPAGATION

This section presents the principles of Alternating back-propagation (ABP) to train a generative model, as proposed in (Han et al., 2017) and presented in (Challu et al., 2022b), and the connection with our proposed latent vector inference.

Let $\mathbf{y} \in \mathbb{R}^D$ be a $D$-dimension data vector (such as an image or time-series window) and $f$ be a generative model with parameters $\boldsymbol{\theta}$,

$$
\begin{aligned}
\mathbf{y} &= f(\mathbf{z}, \boldsymbol{\theta}) + \boldsymbol{\epsilon} \\
\mathbf{z} &\sim N(0, \boldsymbol{I}_d), \boldsymbol{\epsilon} \sim N(0, \sigma^2 \boldsymbol{I}_D)
\end{aligned}
\tag{10}
$$

where $\mathbf{z} \in \mathbb{R}^d$ are latent factors and $d < D$. Let $\{\mathbf{y}^{(i)}, i = 1, ..., n\}$ be $n$ training observations. In Alternating Back-Propagation, parameters $\boldsymbol{\theta}$ are trained by maximizing the observed log-likelihood

$$
L(\boldsymbol{\theta}) = \sum_{i=1}^{n} \log p_{\boldsymbol{\theta}}(\mathbf{y}^{(i)}) = \sum_{i=1}^{n} \log \int p_{\boldsymbol{\theta}}(\mathbf{y}^{(i)}, \mathbf{z}^{(i)}) d\mathbf{z}^{(i)}
\tag{11}
$$

The gradients $L'(\boldsymbol{\theta})$ for observation $i$ are given by,

$$
\frac{\partial}{\partial \boldsymbol{\theta}} \log p_{\boldsymbol{\theta}}(\mathbf{y}^{(i)}) = \mathbb{E}_{p_{\boldsymbol{\theta}}(\mathbf{z}^{(i)}|\mathbf{y}^{(i)})} \left[ \frac{\partial}{\partial \boldsymbol{\theta}} \log p_{\boldsymbol{\theta}}(\mathbf{y}^{(i)}, \mathbf{z}^{(i)}) \right]
\tag{12}
$$

where $p_{\boldsymbol{\theta}}(\mathbf{z}^{(i)}|\mathbf{y}^{(i)})$ is the posterior distribution. ABP approximates this intractable expectation with Monte Carlo average by taking a single sample $z^*$ from the posterior distribution with approximate Langevin Dynamics, by iterating

$$
\mathbf{z}_{t+1}^{(i)} = \mathbf{z}_t^{(i)} + \frac{s}{\sigma_z} \frac{\partial}{\partial \mathbf{z}^{(i)}} \log p_{\boldsymbol{\theta}}(\mathbf{z}_t^{(i)}|\mathbf{y}^{(i)}) + \sqrt{2s}\boldsymbol{\epsilon}_t
\tag{13}
$$

where $\boldsymbol{\epsilon}_t \sim N(0, \boldsymbol{I}_D)$, $s$ is the step size, and $\sigma_z$ controls the annealing or tempering. Finally, the Monte Carlo approximation of the gradient is given by

$$
L'(\boldsymbol{\theta}) \approx \frac{\partial}{\partial \boldsymbol{\theta}} \log p_{\boldsymbol{\theta}}(\mathbf{z}^{*(i)}, \mathbf{y}^{(i)})
\tag{14}
$$

For our proposed latent vector inference, we reformulate the posterior sampling as a non-convex minimization problem presented in 3, which aims to minimize the mean square error (MSE) between the reconstruction and ground truth $\mathbf{y}^{(i)}$. Minimizing the MSE is equivalent to the maximum a posteriori estimation (MAP) of $p_{\boldsymbol{\theta}}(\mathbf{z}^{(i)}|\mathbf{y}^{(i)})$ assuming Gaussian distributions as in equation 10. This methodology also allows for using more sophisticated gradient-based methods than equation 13, such as Adam, can be used to improve the convergence.

## A.3 BENCHMARK DATASETS

Table 5 presents summary information for the benchmark datasets, including the granularity (frequency), number of features and timestamps, and train/validation/test proportions.

Table 5: Summary of benchmark datasets

| DATASET | GRANULARITY | # OF FEATURES | # OF TIMESTAMPS | TRAIN/VAL/TEST SPLIT |
|---------|-------------|---------------|-----------------|----------------------|
| SIMULATED7 | NA | 7 | 20,000 | 80/10/10 |
| ILI | WEEKLY | 7 | 966 | 70/10/20 |
| ETTM2 | 15 MINUTE | 7 | 57,600 | 60/20/20 |
| EXCHANGE | DAILY | 8 | 7,588 | 70/20/10 |
| SOLAR | HOURLY | 32 | 8,760 | 60/20/20 |
| WEATHER | 10 MINUTE | 21 | 52,695 | 70/10/20 |

All datasets are public, and are available in the following links:

- **ILI**: https://gis.cdc.gov/grasp/fluview/fluportaldashboard.html
- **Exchange**: https://github.com/laiguokun/multivariate-time-series-data
- **Solar**: https://www.nrel.gov/grid/solar-power-data.html
- **ETTm2**: https://github.com/zhouhaoyi/ETDataset
- **Weather**: https://www.bgc-jena.mpg.de/wetter/

## A.4 SIMULATED7 DATASET

**Simulated7** is a synthetic dataset that we design to evaluate forecasting models robustness to missing data and distribution shifts in a controlled environment. It consists of 7 time-series, each generated independently as the sum of two cosine functions with different frequencies and a small Gaussian noise. In particular, it is composed as the sum of the three following elements:

$$
\begin{aligned}
y_t^{\text{low}} &= \cos(it) & , i \sim U(5, 50) \quad t \in [0, 5] \\
y_t^{\text{high}} &= \cos(it) & , i \sim U(100, 300) \quad t \in [0, 5] \\
y_t^{\text{noise}} &\sim N(0, 0.001) & , t \in [0, 5]
\end{aligned}
\tag{15}
$$

Each time-series consists of 20,000 timestamps, with regular intervals between $[0, 5]$. For the *missing data* we obfuscate random timestamps following the procedure described in section 4, using $s = 10$ and $p_o \in \{0, 0.2, 0.4, 0.6, 0.8\}$. For simulating *distribution shifts* we perturb **only the test set** with two transformations: adding a linear *trend* with slope 6, and scaling the *magnitude* by 0.5.

## A.5 HYPERPARAMETER OPTIMIZATION

`SpectraNet` hyperparameters are tuned on the validation set of each dataset using HYPEROPT algorithm with 30 iterations, Table 6 presents the hyperparameter grid. To ensure a fair comparison with baseline models, we also tuned their respective hyperparameters with the same procedure, as the default configuration in their implementations might not perform well in the different settings we explore. We detail the hyperparameter grid for each baseline model next.

Table 6: Hyperparameters grid for `SpectraNet`.

| HYPERPARAMETER | VALUES |
|---|---|
| Learning rate | $\{0.0001, 0.0005, 0.001, 0.005\}$ |
| Training steps | $\{500, 1000\}$ |
| Batch size | $\{8, 16, 32\}$ |
| Random seed | $[1,10]$ |
| GD iterations during training | $\{25, 50\}$ |
| GD iterations during inference | $\{300, 1000\}$ |
| GD step size | $\{0.2, 1, 2\}$ |
| Max degree trend polynomial | $\{3\}$ |
| Window size ($s_w$) | $\{128, 256, 512\}$ |
| Kernel size | $\{4, 8\}$ |
| Stride | $\{4\}$ |
| Dilation | $\{1\}$ |
| Convolution filters of two hidden layers ($n_1$,$n_2$) | $\{(256,128), (128, 64)\}$ |

For N-HiTS, we use the hyperparameter grid described in their paper (Challu et al., 2022a). The N-HiTS is a generalization of the N-BEATS, which is already included in the hyperparameter grid as a posible configuration. For the Informer and Autoformer we explore different values for the dropout probability, number of heads, learning rate (same as `SpectraNet`), batch size (same as `SpectraNet`), size of embedding, and sequence length. For StemGNN we explore different optimization parameters, including the learning rate, batch size, and epochs. Finally, for RNN we explore different dilations, number of layers, hidden size, and optimization parameters. For the N-HiTS, transformers, and RNN models we used the `Neuralforecast` library (available in PyPI and Conda).

## A.6 SPECTRANET TRAINING ALGORITHM

Algorithm 1 presents the training procedure of `SpectraNet`. The model is trained for a fixed number of iterations $n_{iters}$, randomly sampling $b$ windows in each iteration. Parameters $\boldsymbol{\theta}$ are optimized with Adam optimizer.

---

**Algorithm 1:** Training procedure

---

**input** : multivariate time-series $\mathbf{Y} \in \mathbb{R}^{m \times T}$, model $F_{\boldsymbol{\theta}}$, learning iterations $n_{iters}$
**output:** $F_{\boldsymbol{\theta}^*}$, inferred latent vectors $\{\mathbf{z}_t^*, t = 0, ..., T\}$
Let $i \leftarrow 0$, initialize $\boldsymbol{\theta}$
Initialize $\mathbf{z}_t$, for $t = 0, ..., T$
**while** $i < n_{iters}$ **do**

    Take a random mini-batch of windows $\{\mathbf{Y}_{j_k - L : j_k + H}, j_k \sim U(L, T - H), k\{1, ..., b\}$.
    $\mathbf{z}_{j_k}^* \leftarrow \arg\min_{\mathbf{z}} L(\mathbf{Y}_{j_k - L : j_k}, \hat{\mathbf{Y}}_{j_k - L : j_k}(\mathbf{z})), k \in \{1, ..., b\}$
    Update $\boldsymbol{\theta}$ with Adam using $\mathbf{z}^*$ as input.
    Store $\mathbf{z}_{j_k}^*, k \in \{1, ..., b\}$
    $i \leftarrow i + 1$
**end**

---

## A.7 FORECASTS ON SIMULATED7

Figure 1 presents the complete forecasts on the test set for `SpectraNet`, N-HiTS, Informer, and RNN models. `SpectraNet` is the only model that can accurately forecast with up to 80% missing data and changes in distribution.

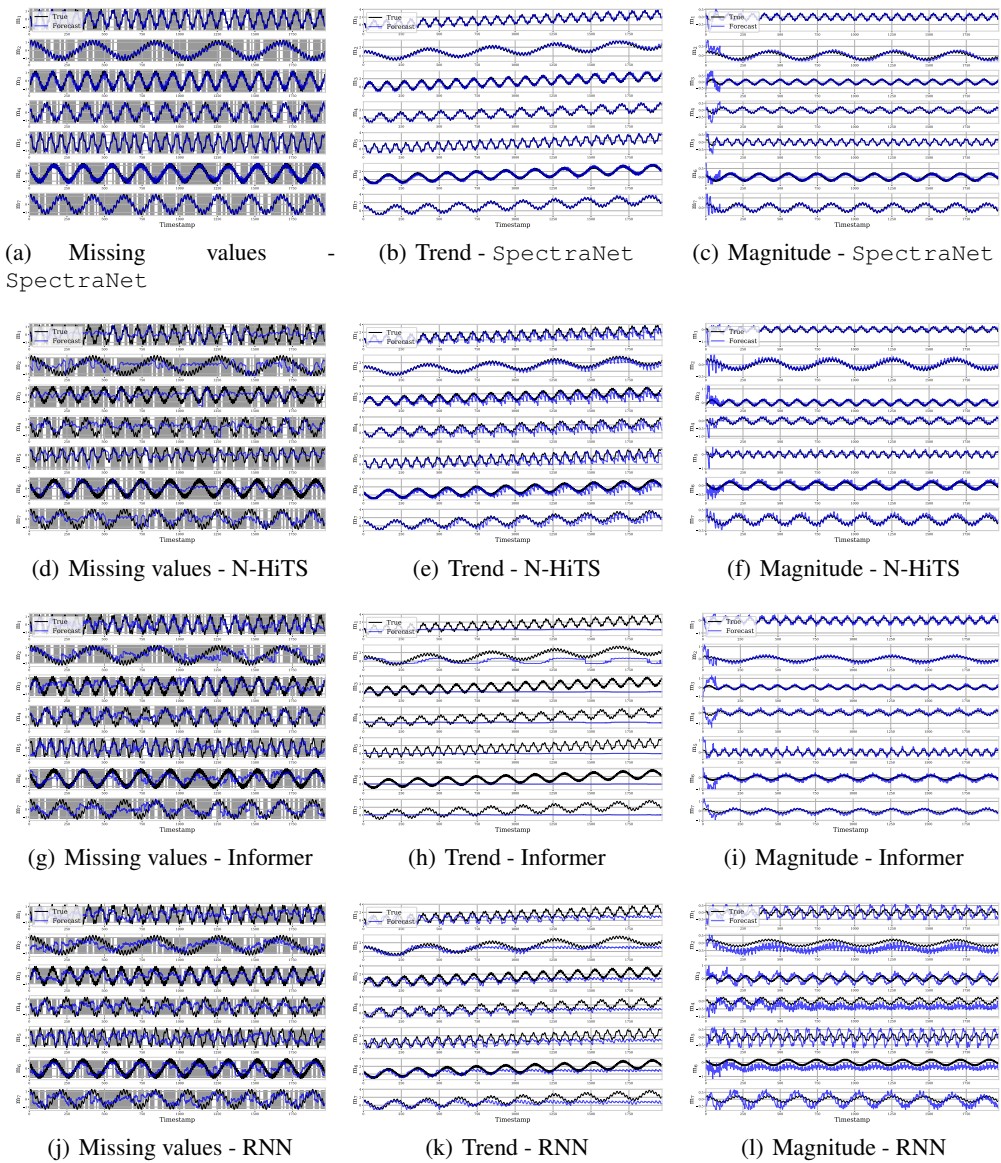

(a) Missing values - `SpectraNet`

(b) Trend - `SpectraNet`

(c) Magnitude - `SpectraNet`

(d) Missing values - N-HiTS

(e) Trend - N-HiTS

(f) Magnitude - N-HiTS

(g) Missing values - Informer

(h) Trend - Informer

(i) Magnitude - Informer

(j) Missing values - RNN

(k) Trend - RNN

(l) Magnitude - RNN

Figure 1: Forecasts on the test set for all features of **Simulated7** dataset with 80% of missing data (missing regions in grey), change in trend, and change in magnitude. Forecasts are produced every 24 timestamps in a rolling window strategy.

## A.8 ILI DATASET

Figure 2 shows the time series of the ILI dataset and the start of the test set (vertical red line). The first five features present considerably larger and longer spikes on the test set. This is a particularly challenging task, as most models tend to underestimate the larger spikes.

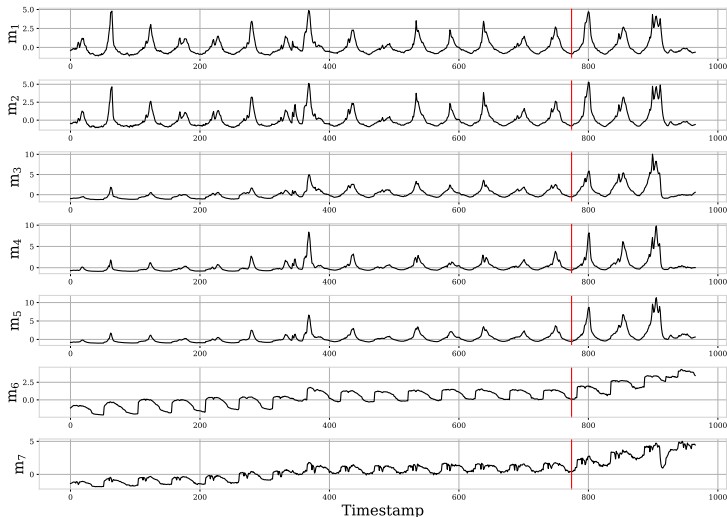

Figure 2: ILI dataset, start of the test set marked with vertical red line.

## A.9 ADDITIONAL OCCLUSION EXPERIMENTS

*In this section we present additional occlusion experiments to analyze the case where the missing data regime (how much or for how long missing data occurs) differs between the training and test sets. In this setting we train models on complete data and inject missing values only during inference on the test set. The following table presents SpectraNet and N-HiTS (best baseline) results on the Simulated7 dataset. SpectraNet performance is almost identical to the original results, demonstrating the method's robustness to changes in the behavior of the presence of missing data.*

Table 7: Forecasting accuracy on **Simulated7** dataset with missing values only on the test set, forecasting horizon of 24 timestamps. Lower scores are better. Metrics are averaged over five runs, best model highlighted in bold.

| $p_o$ | SpectraNet MSE | SpectraNet MAE | N-HiTS MSE | N-HiTS MAE |
|---|---|---|---|---|
| 0.2 | **0.008** | **0.079** | 0.013 | 0.146 |
| 0.4 | **0.017** | **0.085** | 0.031 | 0.224 |
| 0.6 | **0.021** | **0.097** | 0.257 | 0.384 |
| 0.8 | **0.029** | **0.124** | 0.463 | 0.562 |

## A.10 CNN ABLATION STUDIES

Figure 1 of the paper presents the default recommended configuration for `SpectraNet`. In this section we explore different configurations for the CNN, in particular for the number of hidden layers. For the model SpectraNet$_{i,j}$, $i$ refers to the number of layers with temporal resolution $s_w/2$, and $j$ to the number of layers with $s_w$ resolution. The following table presents the results on ILI and Solar datasets. The results suggest SpectraNet's performance is not significantly affected by the number of layers in these datasets for several occlusion probabilities. On average, SpectraNet$_{2,2}$ achieves the best performance.

Table 8: Forecasting accuracy results of `SpectraNet` with different CNN architectures on benchmark datasets with different proportion of missing values ($p_o$), forecasting horizon of 24 timestamps, lower scores are better. Metrics are averaged over five runs, best model highlighted in bold.

|  | $p_o$ | SpectraNet MSE | SpectraNet MAE | SpectraNet$_{2,2}$ MSE | SpectraNet$_{2,2}$ MAE | SpectraNet$_{3,2}$ MSE | SpectraNet$_{3,2}$ MAE | SpectraNet$_{3,3}$ MSE | SpectraNet$_{3,3}$ MAE |
|---|---|---|---|---|---|---|---|---|---|
| ILI | 0.0 | 0.724 | 0.557 | 0.719 | 0.549 | 0.773 | 0.560 | 0.726 | 0.550 |
| ILI | 0.2 | 1.153 | 0.662 | 1.148 | 0.670 | 1.164 | 0.682 | 1.156 | 0.658 |
| ILI | 0.6 | 2.453 | 1.012 | 2.391 | 1.002 | 2.382 | 0.985 | 2.369 | 0.991 |
| Solar | 0.0 | 0.007 | 0.054 | 0.007 | 0.052 | 0.008 | 0.058 | 0.007 | 0.053 |
| Solar | 0.2 | 0.012 | 0.058 | 0.010 | 0.059 | 0.012 | 0.059 | 0.011 | 0.054 |
| Solar | 0.6 | 0.013 | 0.068 | 0.012 | 0.064 | 0.014 | 0.066 | 0.015 | 0.064 |

