# OpenReview forum: "SpectraNet: multivariate forecasting and imputation under distribution shifts and missing data"
_ICLR.cc/2023/Conference — Submitted to ICLR 2023_

### Official Review · Reviewer_r8BM · 2022-10-19

**Confidence:** 4
**Correctness:** 3
**Technical Novelty And Significance:** 3
**Empirical Novelty And Significance:** 4
**Recommendation:** 8

**Clarity, Quality, Novelty And Reproducibility:**

Clarity: Good
Quality: Good
Novelty: Good
Reproducibility:Good

**Strength And Weaknesses:**

Pros:
1) The idea to optimize a latent vector for different data is interesting and this is the first time this idea is applied on time series. Previously this idea has been used to images [2].
2) Experiments result are better than several baselines

Cons:
1) The idea to combine several basis functions to reconstruct the input is not new. Previously, [1,3] used this idea to reconstruct music signals. Also, I am not sure that only using sin, cos, and poly functions are enough for all types of time series datasets. Some datasets may have strange shapes that can not be reconstructed well.
2) For optimization, I am not sure about using the inference step and the learning step can lead to a globally optimal solution. It would be great if the authors can provide some explanation here.

[1] DDSP: Differentiable Digital Signal Processing
[2] Optimizing the Latent Space of Generative Networks
[3] Differentiable Wavetable Synthesis

**Summary Of The Paper:**

To tackle the distribution shift and missing data problem of time series, this paper proposes a latent space spectral decomposition method for simultaneous time series forecasting and imputation. The latent vector is optimized individually for unseen data and thus can generalize well to unseen data distribution.

**Summary Of The Review:**

Despite the drawbacks that I mentioned, I think this is an interesting paper overall.

---

> ### Author Response · Authors · 2022-11-15
> **Response**
>
> We thank the reviewer for the recognition of the paper's contribution.
>
> We thank the reviewer for the additional references on the basis functions; we will add them in the following revised version. Indeed, as we mentioned in the paper, some prior work already used pre-defined basis functions within their architecture. However, one key difference between previous work (including [1], N-BEATS) and ours is that we used the basis as a latent representation. It serves several essential purposes: (1) enforce temporal relations across the reconstructed window (ablation studies show how simply feeding z into the CNN does not produce optimal results), (2) encode shared temporal dynamics for all features (useful to forecast features with missing data), and (3) produce an initial forecast, which is then refined with the CNN. The last point is crucial and solves the potential weakness mentioned by the reviewer. The convolutional filters and non-linearities of the CNN sequentially refine the temporal embedding and can generate more complex forecasts, as demonstrated in some noisy benchmark datasets such as Exchange.
>
> Indeed, for optimization, gradient descent on non-convex objectives is not guaranteed to converge to the global optima, but it is guaranteed to converge to a local optimum (under some conditions). However, this is true for all Deep Learning based methods, and it is not a weakness specific to our model. Following good practices in the literature, we repeated the experiments several times to compare the model's average performance across different runs.

---

> > ### Comment · Reviewer_r8BM · 2022-12-02
> > **Thanks for your clarification**
> >
> > I have read the comments from other reviewers, and I still believe this paper is a novel approach to time series data analysis despite some minor drawbacks. Firstly using a latent basis is novel. Secondly, the method is novel because, in contrast to previous models such as GAN, CNN, and Transformer for time series analysis, this paper introduces a model that's more interpretable.

---

### Official Review · Reviewer_HyvP · 2022-10-23

**Confidence:** 5
**Correctness:** 3
**Technical Novelty And Significance:** 2
**Empirical Novelty And Significance:** 2
**Recommendation:** 5

**Clarity, Quality, Novelty And Reproducibility:**

Details are in [Weaknesses]

**Clarity**:
The clarity is good but with some minor mistakes.

**Quality**:
Clear-written and well-organized.

**Novelty**:
Motivation is good but the contributions are not significant.

**Reproducibility**:
The training procedure is clear in Appendix 5,  and authors promise to release the code after paper acceptance.


**Strength And Weaknesses:**

**Strengths**:
1. Well-organized and clearly written.
2. Promising results.


**Weaknesses**:
1.  In Sec. 3.5, the distribution shift is defined as the difference of missing data regime between training set and test set. However, in my opinion, this kind of difference is not the distribution shift, since the training and test sets both come from the **same** dataset in a random split (or not) manner, the **data distribution** should be the same in the training and test sets, even though the total numbers or locations of missing values are different. Therefore, I would say there is no distribution shift problem.

2. Some key details are missing.
(a) In Sec. 3.2, how many temporal bases $B$ are used? What's the insights or reasons?
(b) In Sec. 3.3, there is no clarification of the ConvTranspose1d, more details should be included. Why do the first two layers learn a common representation and the second layer refine the temporal resolution? There is no such experiment or explanation of this claim. In addition, the *causality* is not clear either, "the latent vector $z^{*}$ is only inferred with information on the reference window" is not the reason of  *causality*.
(c) In Sec. 4.3, it says "all models are trained with the training loss only on available values", how about the missing points? I think the ground truth of missing points is accessible during training, if the model doesn't use the missing points during training, what's the meaning of interpolation?



3. Experiments are not sufficient. There is no ablation study to demonstrate the effectiveness of some designs in the model, i.e., temporal basis, the inference step, etc.


4. Minor mistakes:
(1) In Sec. 3.1, the first sentence is incoherent, "of the", "instead of".
(2) In Fig. 2, the legend blocks the first curve $\hat{y_{1}}$,  even though it looks the same as $\hat{y_{10}}$ . Almost the same problem in all the figures.
(3) Eq. (8), not aligned.
(4) In Fig.3, the characters are too small.
(5) In References, the names of conferences or journals, i.e. AAAI or AAAI Press , are not consistent.
I highly encourage authors to revise the submission to avoid such small mistakes, as well as some typos.


5. Some missing references[1,2] when addressing distribution shift problem in time-series.
[1] Adaptive Trajectory Prediction via Transferable GNN. CVPR2022
[2] Expanding the Deployment Envelope of Behavior Prediction via Adaptive Meta-Learning. arXiv 2022





**Summary Of The Paper:**

In this paper, authors proposed a multivariate time-series forecasting model (SpectraNet) that unified the forecasting and interpolation problem. Specifically, the model first infers the optimal latent vector on the reference window by minimizing the reconstruction error, then the model generates the full predictions via a latent space spectral decomposition module. The proposed model achieves the best performance compared with other existing methods.


**Summary Of The Review:**

I list my concerns in [Weaknesses], I am happy to discuss and increase the rating if my concerns are addressed.

---

> ### Author Response · Authors · 2022-11-14
> **Response (2/2)**
>
> Regarding missing key details:
> - The size of the basis is given by equation 7 and depends on the max degree of the polynomial and window size $(p+1 + s_w/2 + s_w/2)$, both defined in Table 6 of the Appendix; we will explicitly state the final size of the basis in the next revised version.
> - We use the term "ConvTranspose1d" based on PyTorch and Tensorflow names for the layer; we acknowledge it can be confusing. We briefly defined it in Section 3.3 and will explain it in more detail in the next revised version.
> - The Convolution Network (CNN) presented in Section 3.3 is our default recommended architecture for SpectraNet, similar to how baseline models fixed some aspects of their architecture. Following classic convolutional decoders in computer vision, the Top-Down CNN of SpectraNet increases the resolution of the output through the layers. Our experiments demonstrate how the Top-Down CNN with only three layers can achieve SoTA performance in a wide range of benchmark datasets. However, the number of layers is a tunable hyperparameter; larger and deeper networks could be used for larger datasets. We ran additional ablation studies on the number of layers of the CNN. The new table 8 on Appendix A.10 of the latest submission presents the new results on ILI and Solar datasets. The results suggest SpectraNet's performance is not significantly affected by the number of layers in these datasets.
> - Following the definition in the seminal paper "An Empirical Evaluation of Generic Convolutional and Recurrent Networks for Sequence Modeling" on Temporal Convolution Networks (TCN), causality refers to no information leakage from the future to the past. Causality is a crucial property of models, as forecasts can only be produced with currently available information. The inference of the latent vectors uses only the ground truth of the "reference window" to forecast the "forecasting window," so there is no information leakage from the future (i.e., is causal). In this paragraph, we want to highlight the difference between our model and the "causal convolutions" of a classic TCN.
> - The ground truth of missing points is not used while training because the fundamental idea of these experiments is to demonstrate how the models perform under missing data on both training and test data. Injecting missing points in initially complete datasets allows us to create a controlled setting and vary the proportion of missing data. The difference with the interpolation task is how models are evaluated. For the interpolation task, models are only evaluated on their ability to recover past (not future) missing points. The missing data cannot be used as inputs or labels for training. Finally, the new results in Table 7 of Appendix A9 show when the training data is complete.

---

> ### Author Response · Authors · 2022-11-18
> **Response (1/2)**
>
> We thank the reviewer for the positive recognition of the experimental results and clarity of the paper and for the insightful comments that helped us improve the paper. **The latest version of the paper (see PDF) includes the changes mentioned below**.
>
> ### Distribution shift
> Distribution shifts are a well-known problem in time series forecasting. We define distribution shifts in the third paragraph of the introduction as "discrepancies in distribution between the train and test data" and provide examples in our simulated setting (Figure 3, panels b and c) and the benchmark ILI dataset (Figure 2 of the Appendix). This definition refers to the actual time series (ground truth).
>
> Distribution shift arises due to the forecasting task's nature, as the time series' underlying generating process can change over time. In time series forecasting, the standard approach is not to randomly split the train and test sets but in two exclusive sets chronologically ordered with a "cutoff" timestamp, which resembles how forecasting models are used in practice. In Section 3.5, we mentioned that our method is additionally robust to changes in the behavior of the presence of missing data over time, which is a separate issue from distribution shifts. One example of this case is when the training data is complete, and timestamps are only missing during inference in the test set. This is a common problem in some domains, such as healthcare and IoT. Sensors can fail or disconnect unexpectedly on production/inference after the model is trained on complete data.
>
> We run an additional experiment to support this claim, where we train models on complete data and inject missing values only during inference on the test set. The new table 7 on Appendix A.9 of the latest submission presents the new results on the Simulated7 dataset. As claimed, SpectraNet performance is almost identical to the original results, demonstrating the method's robustness to changes in the behavior of the presence of missing data. N-HiTS performance is worst than the initial results, as the model always expects complete data (training with missing data is similar to the effect of using dropout on the input layer).
>
> ### Ablation studies
> Table 4 in Section 4.4 of the paper already presents the two ablation studies mentioned by the reviewer to demonstrate the contribution of both using a temporal basis and inference of latent vectors. We compared SpectraNet with two different versions of the model: SpectraNet1 uses a CNN encoder instead of inferring the latent vector, and SpectraNet2 maintains the inference step but removes the temporal basis. We will include diagrams of both SpectraNet1 and SpectraNet2 in the appendix for the revised version.
>
> ### References
> We thank the reviewer for the additional references on distribution shits. We did not initially include them as they were made public very close to or after our submission; we already added them in the latest submission.
>
> ### Novelty
> Regarding novelty, the paper makes both significant methodological and empirical contributions. First, SpectraNet encompasses a new forecasting approach with a substantially different core idea to the most currently studied methods (Transformers, Recurrent models, MLPs, GNNs). The technique of inferring latent vectors can have multiple other applications, such as transfer learning. On the empirical side, SpectraNet achieves SoTA performance in forecasting and imputation tasks and makes strong progress in tackling several understudied and widespread challenges.

---

### Official Review · Reviewer_Mu1M · 2022-10-25

**Confidence:** 4
**Correctness:** 2
**Technical Novelty And Significance:** 2
**Empirical Novelty And Significance:** 2
**Recommendation:** 3

**Clarity, Quality, Novelty And Reproducibility:**

This paper is difficult to follow due to the unclear presentation, especially in the model part. Further improvement is needed to improve the clarity.

**Strength And Weaknesses:**

Strengths
1. An important problem is studied
Weaknesses:
1. Weak motivation for proposed model components
2. Paper presentation and organization need to be improved

**Summary Of The Paper:**

In the paper, the authors proposed a time-series forecasting framework to overcome the distribution shifts and tackle missing values simultaneously.

**Summary Of The Review:**

In this paper, the authors proposed a novel time series forecasting framework that handles missing data and distribution drift. However, the paper is not clearly presented, and the following issues should be addressed:

1. The presentation and organization of this paper need to be improved. There are many typos and incomplete sentences. For example, on page 2, "practice, imputation models are first ...".

2. What's the advantage of inferring the latent variable separately instead of using an encoder? The motivation is not strong here, and the authors didn't provide enough details. Besides, a point-wise distance-based loss function (e.g., MSE) may not be enough to capture the shape correlation between two-time series. Can the authors provide more information and analysis regarding the design of latent vector inference?

3. What's the meaning of s_w / 2 in Figure 1 and equation 7? What is p in equation 7? The authors should clearly describe and explain the equations used in the paper.

4. How does the imputation work in the inference of latent vectors? Can the authors provide more details?

---

> ### Author Response · Authors · 2022-11-16
> **Response (2/2)**
>
> ### MSE
> SpectraNet uses the Mean Square Error (MSE) as reconstruction loss for several reasons:
> - MSE is one of the most used metrics for reconstruction error, even on structured data such as time-series windows or images. For example, it is used by many multivariate time-series anomaly detection reconstruction-based methods [1] to compute the anomaly scores, also by VAE as the reconstruction loss, and as content loss for image neural style transfer.
> - It is theoretically founded as minimizing MSE is equivalent to maximizing a Gaussian likelihood.
> - MSE is differentiable and fast to compute.
> - Figure 2 shows how by minimizing the MSE in the reference window, SpectraNet can accurately predict future observations. The SoTA performance of SpectraNet is further demonstrated in Tables 1, 2, and 3.
>
> Using more complex distance metrics that consider temporal correlations might further improve the performance of SpectraNet and is an interesting line for future research.
>
> [1] Challu, Cristian I., et al. "Deep Generative model with Hierarchical Latent Factors for Time Series Anomaly Detection." International Conference on Artificial Intelligence and Statistics. PMLR, 2022.
> ### Equation 7
> The term $s_w/2$ means $s_w$ (size of the window) divided by two; we will change it to $\frac{s_w}{2}$. $p$ is a user-defined hyperparameter to control the maximum degree of the polynomial trend basis; we will add this explanation after equation 7.
>
> ### Imputation
> An imputation model aims to recover missing observations using the observed values. Let $\mathbf{Y} \in \mathbb{R}^{m,s_w}$ be a time series window with $m$ features and $s_w$ timestamps and let $\mathbf{I_o} \in \mathbb{R}^{m,s_w}$ be a 1-0 mask matrix indicating which data points are available in window $\mathbf{Y}$. SpectraNet can be used as an imputation model since the Convolution Network (CNN) outputs the complete window $\hat{\mathbf{Y}}$ (including missing data) from the latent vector $\mathbf{z}$. In this case, the latent vector is inferred by minimizing $L(\mathbf{I_o} \circ \hat{\mathbf{Y}}(\mathbf{z}), \mathbf{I_o} \circ \mathbf{Y} )$, where $\circ$ is the element-wise multiplication. That it, the reconstruction error only considers the available observations. Our results in Table 3 demonstrate that SpectraNet achieves SoTA imputation performance on several benchmark datasets, improving over the latest Deep Learning imputation models.
>
> ### Novelty and contribution
> Regarding novelty, the paper makes both significant methodological and empirical contributions. First, SpectraNet encompasses a new multivariate forecasting approach with a substantially different core idea to the most currently studied methods (Transformers, Recurrent models, MLPs, GNNs). The technique of inferring latent vectors can have multiple other applications, such as transfer learning. On the empirical side, SpectraNet achieves SoTA performance in forecasting and imputation tasks and makes substantial progress in tackling several understudied and widespread challenges.

---

> ### Author Response · Authors · 2022-11-16
> **Response (1/2)**
>
> We thank the reviewer for the insightful comments that helped us improve the model's presentation and motivation. **The latest version of the paper (see PDF) includes the changes mentioned below**.
>
> **Based on the reviewer's suggestion, we: (1) formally present the latent vector inference algorithm in the context of the Alternating back-propagation framework; (2) merge parts of section 3.5 into 3.1; (3) state the advantages of mean square error, (4) improve the explanation of equation 7, and (5) explain how latent vectors are inferred when information is incomplete. The details are explained below.**
>
> ### Presentation
> The sentence "practice, imputation models are first ..." is not incomplete; it starts at the end of the previous page. We acknowledge this might be confusing to read; we will add line breaks to improve the presentation and avoid splitting sentences between pages.
>
> ### Inference of latent vectors
> We thank the reviewer for the comments on the motivation of the latent vector inference. The main advantage of SpectraNet is the robustness to missing data and distribution shifts, two widespread challenges of real-world applications which can significantly affect the performance of SoTA forecasting models. Designing models capable of addressing these challenges can have significant benefits in multiple domains, including high stake areas such as healthcare.
>
> In the original submission, Equation 3 presents the optimization problem behind latent vector inference, and Section 3.1 explains how the optimization problem is solved. Section 3.5 explains why the inference of latent vectors is a crucial architecture design for the model's robustness to missing data and distribution shifts. Figure 2 presents an example of how the model's output evolves during inference.
>
> Our proposed method is based on the Alternating back-propagation (ABP) algorithm (Han et al., 2017), as mentioned in the Related Work appendix. This work demonstrated that generative models trained with ABP achieved superior performance than VAE (which uses encoders) on recovering missing segments of images. Instead of using an encoder, ABP samples the latent vector from the posterior distribution with Langevin Dynamics, and the models are trained by maximizing the observed likelihood directly. In the ABP framework, our latent vector inference is the maximum a posteriori estimation (MAP) of the posterior distribution, $P(\mathbf{Z}|\mathbf{Y})$.
>
> Intuitively, the inference step adds an extra degree of freedom for the model to find the optimal latent vector that best fits the current dynamics of the window based only on the available information (see below in the Imputation question for more details). On the contrary, once trained, encoders are a fixed parametric function $F:Y\rightarrow Z$, which heavily relies on all the inputs. We empirically demonstrate the significant performance gains of the inference step in the main results (compared to other architectures), and in the ablation studies, by comparing SpectraNet to an alternative architecture that uses an Encoder.

---

### Author Response · Authors · 2022-11-18
**Summary of changes**

We thank the reviewers for their valuable comments that helped us improve the paper's presentation, motivation, and experimental results. In this response, we summarize the changes to the revised version based on the reviewers' questions and suggestions. To facilitate the review of the updated version, we highlighted the differences in blue in the PDF. The main changes include:

- **Latent vector inference details and motivation**: improved motivation for latent vector inference, formally presenting the proposed algorithm in the context of the Alternating back-propagation framework. Merged parts of 3.5 into 3.1 to highlight the advantages of latent vector inference. Explained benefits of mean square error (MSE) as reconstruction loss. Explained how latent vectors are inferred with incomplete data. See Section 3.1 and Appendix A.2.
- **LSSD**: improved clarity of equation 8 (previously equation 7) and hyperparameter $p$, added the total number of basis functions. See Section 3.2.
- **Convolution Neural Network**: renamed ConvTranspose1d, improved explanation of causality, and the number of layers. See Section 3.3
- **New sets of results**: (1) occlusion experiment with different missing probabilities $p_o$ between train and test sets, and (2) ablation study on the number of layers of the CNN. See Appendices A.9 and A.10.
- **New references**: distribution shifts and use of basis functions.
- **Minor fixes**: references, line breaks, typos. We are working on improving the plots.

We kindly ask reviewers to consider improving the scores based on the improvements to the paper and the responses below.

---

### Decision · Program_Chairs · 2023-01-20

**Decision:**

Reject

**Justification For Why Not Higher Score:**

- The idea to combine several basis functions to reconstruct the input is not new.
- Gradient descent on non-convex objectives could not guarantee to converge to the global optima.
- The comparison to the previous work may not be fair.


**Justification For Why Not Lower Score:**

N/A

**Metareview: Summary, Strengths And Weaknesses:**

The paper proposes SpectraNet that can simultaneously produce forecasts and interpolate past observations. The method achieves some good overall performance.

Two reviewers do not support the publication of this paper at ICLR. After reviewing and considering carefully, I also think that the paper is not ready for the publication.

1) The idea to combine several basis functions to reconstruct the input is not new. The authors explain that this paper uses the basis as a latent representation which is different. However, this is not really significant novel in terms of contributions.
2) The authors mentioned that for optimization gradient descent on non-convex objectives could guarantee to converge to the global optima. This is generally not true since there is no rigorous work that can show convergence to global optimal solution in the deep learning applications.
3) The comparison to the previous work may not be fair. The authors did not mention about training epochs (I see 500 and 1000 in appendix) and I wonder if it is a fair comparison with the previous work. In Autoformer paper, their training process is early stopped within 10 epochs.
4) The experimental results seem skeptical since the authors do not provide the comparison with some latest state-of-the-art method such as in Zeng et al. “Are Transformers Effective for Time Series Forecasting?”. Moreover, the numbers in the tables seem quite different.
5) There is no code to reproduce results. Therefore, it is hard to justify the effectiveness of the methods.

The results may be promising but not clear if it is a fair comparison or not.